# Verification of the role of exosomal microRNA in colorectal tumorigenesis using human colorectal cancer cell lines

**Gyoung Tae Noh[1], Jiyun Kwon[2], Jungwoo Kim[2], Minhwa Park[2], Da-Won Choi[2], Kyung-Ah Cho[2], So-Youn Woo[2], Bo-Young Oh[3], Kang Young Lee[1], Ryung-Ah Lee[4]***

**1** Department of Surgery, Yonsei University College of Medicine, Seoul, South Korea, **2** Department of Microbiology, Ewha Womans University College of Medicine, Seoul, South Korea, **3** Department of Surgery, Hallym University College of Medicine, Seoul, South Korea, **4** Department of Surgery, Ewha Womans University College of Medicine, Seoul, South Korea

* ralee@ewha.ac.kr

**Data Availability Statement:** All relevant data are within the manuscript and its Supporting information files.

## Abstract

Exosomes are a group of small membranous vesicles that are shed into the extracellular environment by tumoral or non-tumoral cells and contribute to cellular communication by delivering micro RNAs (miRNAs). In this study, we aimed to evaluate the role of exosomal miRNAs from colorectal cancer cell lines in tumorigenesis, by affecting cancer-associated fibroblasts (CAFs), which are vital constituents of the tumor microenvironment. To analyze the effect of exosomal miRNA on the tumor microenvironment, migration of the monocytic cell line THP-1 was evaluated via Transwell migration assay using CAFs isolated from colon cancer patients. The migration assay was performed with CAFs ± CCL7-blocking antibody and CAFs that were treated with exosomes isolated from colon cancer cell lines. To identify the associated exosomal miRNAs, miRNA sequencing and quantitative reverse transcription polymerase chain reaction were performed. The migration assay revealed that THP-1 migration was decreased in CCL7-blocking antibody-expressing and exosome-treated CAFs. Colon cancer cell lines contained miRNA let-7d in secreted exosomes targeting the chemokine CCL7. Exosomes from colorectal cancer cell lines affected CCL7 secretion from CAFs, possibly via the miRNA let-7d, and interfered with the migration of CCR2+ monocytic THP-1 cells *in vitro*.

## Introduction

Colorectal cancer (CRC) is one of the most common cancers and accounts for 10% of cancer-related deaths worldwide [1, 2]. From 1990 to 2017, the age-standardized incidence rates of CRC increased globally [3]. On the basis of disability-adjusted life years, CRC has become the fourth leading cause of cancer burden, behind lung cancer, liver cancer, and stomach cancer [3]. Despite the improvements in diagnostic and surgical techniques and introduction of novel chemotherapeutic agents, the prognosis for CRC remains unsatisfactory, with a 13% 5-year survival rate in metastatic CRC due to the heterogeneity of response to therapy [4, 5]. Recently, the tumor microenvironment, including stromal cells, immune cells, and endothelial cells, and

**Funding:** The authors received no specific funding for this work.

**Competing interests:** The authors have declared that no competing interests exist.

**Abbreviations:** CAFs, cancer-associated fibroblasts; DNA, deoxyribonucleic acid; FBS, fetal bovine serum; HRP, horseradish peroxidase; miRNA, micro RNA; PBS, phosphate-buffered saline; qRT-PCR, quantitative reverse transcription-polymerase chain reaction; RNA, ribonucleic acid; SDS-PAGE, sodium dodecyl sulfate polyacrylamide gel electrophoresis.

the communication between its constituents, has emerged as a main factor for therapeutic heterogeneity [6].

In this context, exosomes have attracted increasing attention in the biology of the tumor microenvironment. Exosomes are double-membrane vesicles that are shed into the extracellular environment by tumoral or non-tumoral cells. These vesicles function in cellular communication by exchanging genetic material between cells [7]. Increasing evidence has demonstrated the important roles exosomes play in intercellular communications and their involvement in physiological conditions and pathological processes, such as cancer [8, 9]. Exosomes contain lipids, proteins, and nucleic acid molecules such as DNA, RNA, and viral genomic nucleic acids [10, 11]. Micro RNA acid (miRNA) is one of the exosome cargos and plays a role as a carrier for genetic information, mediating communication between cancer cells [12–14]. Tumor-derived exosomal miRNAs are regarded as putative triggers for malignant transformations. There have been several reports that tumor-derived exosomes contain miRNAs, which can contribute to tumorigenesis by affecting angiogenesis, cell proliferation, migration, and metastasis [15–17].

For CRC, the roles of specific exosomal miRNAs in tumorigenesis have been suggested; they have a significant influence on CRC development and progression, and can also confer the ability to resist anticancer therapy [18–20]. miRNAs such as miR-21, -155, -25-3p, -200b, -210, and -1246 are associated with oncogenic effects such as angiogenesis, migration, invasion, metastasis, and chemoresistance [15–17, 21–26]. miR-196b-5p and miR-142-3p are known to induce cellular stemness, and miR-1249-5p, miR-6737-5p, miR-6819-5p, and miR-10b showed indirect oncogenic activity by modulating the function of fibroblasts [27–29]. In contrast, there are several miRNAs in the miR-96, -149, -486-5p, -6869-5p, -8073, and -193a classes exhibiting tumor suppressor effects [30–34]. Despite accumulating evidence, details regarding the involvement of most miRNAs and the clinical significance of exosomal miRNAs are not yet well defined.

Cancer-associated fibroblasts (CAFs) are vital components of the tumor microenvironment that interact with cancer cells to promote tumorigenesis and progression [6]. In most solid tumors, CAFs are abundant in tumor stroma and play important roles in tumor-stroma communication. CAFs interact with all other cells in the tumor microenvironment and promote tumor progression by secreting various chemokines and cytokines [35, 36]. Diverse CAF-derived factors encourage proliferative signaling in cancer cells to resist cell death and evade growth suppressors [37].

In this study, miRNAs of exosomes from CRC cell lines were profiled by affecting CAFs, and analyzing to investigate their role in tumorigenesis.

## Materials and methods

### Cell culture

HT-29 (American Type Culture Collection, ATCC, Manassas, VA, USA; HTB-38™), SW480 (ATCC, CCL-228™), Jurkat (ATCC, TIB-152™), and THP-1 (Korea Cell Line Bank, Seoul, South Korea) cells were cultured at 37˚C in Roswell Park Memorial Institute medium 1640 (RPMI 1640, WELGENE, Gyeongsan-si, South Korea), including 10% fetal bovine serum (FBS, WELGENE), 100 U/mL penicillin G sodium, and 100 μg/mL streptomycin (Capricorn Scientific, Ebsdorfegrund, Germany). All cell lines were regularly tested using quantitative PCR, Myco-TOOL PCR Mycoplasma Detection Kit (Roche Custom Biotech, Mannheim, Germany). The passage number of HT-29 cells and SW480 cells at purchase were P130 (#lot 700190500 and P99 (#lot 70013709), respectively. Cells were grown to subconfluence (85%–90%) in 10% FBS/ RPMI-1640 media. All cell lines were maintained in 5% $CO_2$ at 37˚C. For exosome isolation, HT-29 and SW480 cells (Table 1) were washed with growth medium using phosphate-buffered

**Table 1. Colorectal cancer cell lines used in this study.**

|  | HT-29 | SW480 |
|---|---|---|
|  | (ATCC HTB-38) | (ATCC CCL-228) |
| Organism | Human | Human |
|  | 44 years, female | 50 years, male |
| Disease | Colorectal adenocarcinoma | Dukes' type B colorectal cancer |
| RAS Mutation | No mutation | Mutation in codon 12 |

saline (PBS), followed by a 10% exosome-depleted FBS (System Biosciences, Palo Alto, CA, USA) supplemented with antibiotics. HT-29 and SW480 cells were cultured with exosome-free FBS for 24 or 48 h, for the isolation of extracellular vesicles.

## Isolation and analysis of exosomes

Exosomes were isolated by ultracentrifugation using an Optima XE-90 equipped with a fixed-angle SW41Ti rotor (Beckman Coulter, Fullerton, CA, USA). First, cell suspensions were centrifuged at 1,500 rpm for 10 min at 4˚C to remove cells and collect the cell supernatant; cell supernatants were then centrifuged at 4,000 rpm for 20 min at 4˚C to remove cell debris, followed by filtering with a 0.22-um syringe filter (Sartorius Stedim Biotech GmbH, Germany) to remove proteins and larger vesicles (i.e., microvesicles). The filtered supernatants were ultracentrifuged at 34,100 rpm for 70 min at 4˚C. The resultant pellets were resuspended in PBS for washing. The final supernatants were ultracentrifuged at 34,100 rpm for 70 min at 4˚C. In addition, exosomes were purified using the ExoQuick-TC™ exosome precipitation solution (System Biosciences, Palo Alto, CA, USA) according to the manufacturer's protocol. The cell supernatant was collected and centrifuged at $300 \times g$ for 15 min at 4˚C to remove the cells and cell debris. The supernatant was transferred to a Macrosep Advance centrifugal device with Omega membrane (Pall, Port Washington, NY, USA) and centrifuged at $5,000 \times g$ for 30 min at 4˚C. Concentrated medium was added to an appropriate volume of ExoQuick solution. After 12 h, the mixture was centrifuged at $1,500 \times g$ for 30 min at 4˚C. The final collected pellets were resuspended in PBS for Nanosight particle tracking analysis (Nanosight NS300, Malvern Instruments Ltd, UK) and in lysis buffer (pro-prep, iNtron Biotechnology, Sungnam-si, South Korea) for immunoblotting. Exosomes were observed using a Hitachi H-7650 transmission electron microscope (Japan). The exosomes were immediately used and stored at -80˚C for up to 1 month.

## Immunoblotting

Cells were plated in a 100-mm culture plate with 8 mL of media. For immunoblotting analysis, 20 µg of protein was resolved with 12% sodium dodecyl sulfate polyacrylamide gel electrophoresis (SDS-PAGE) resolving gel and transferred to polyvinylidene difluoride membranes (GE Healthcare Life science, Pittsburgh, PA, USA). The membranes were probed with anti-CD9 antibody (EPR2949, rabbit monoclonal, ab92726, Abcam, Cambridge, UK), anti-CD63 antibody (MX-49.129.5, mouse monoclonal, ab193349, Abcam), and anti-beta-actin antibody (C4, mouse monoclonal, Santa Cruz Biotechnology, Santa Cruz, CA, USA). The secondary antibodies used were horseradish peroxidase (HRP)-goat anti-rabbit immunoglobulin (IgG) antibody, HRP-conjugated goat anti-mouse IgG F(ab')$_2$ (both from Enzo Life Sciences, Farmingdale, NY, USA), and HRP-conjugated goat anti-mouse IgG (H+L) antibody (#1706516, Bio-Rad Laboratories, Hercules, CA, USA). The images were detected using an ECL chemiluminescent substrate (GE Healthcare Life Science) and analyzed using a LAS-3000 imager (Fujifilm, Japan).

## Isolation and culture of CAFs

Tumor samples from patients undergoing surgery were obtained at Ewha University Medical Center (Seoul, South Korea) in accordance with the ethical guidelines of the institutional review board. Between June 2020 and August 2020, two patients underwent surgical resection for colon cancer were enrolled in this study. The eligibility criteria were a histologically confirmed colonic or rectal adenocarcinoma and major resection of primary lesion. Patients who underwent preoperative treatment such as chemotherapy, radiotherapy, and endoscopic resection were excluded. Before surgery, patients were given an explanation of the purpose and risk of this study and decided to consent to tissue collection. All patients provided their formal, informed, and written consent, agreeing to supply a biopsy for this study. The study and informed consent were reviewed and approved by the institutional review board of Ewha Medical Center Seoul hospital. (IRB No. SEUMC 2019-12-028).

CAFs were isolated from tumor samples of patients. Cancer specimens (8 $mm^2$) were washed three times with PBS, minced into approximately 0.5–1 $mm^2$ pieces, and digested in RPMI-1640 containing 0.05% trypsin (Gibco/Thermo Fisher Scientific, Waltham, MA, USA) and 0.75 mg/mL collagenase type I (Stem Cell Technologies, Vancouver, Canada) for 40 min at 37˚C. The homogenate was collected and passed through a 70-μm pore cell strainer (SPL Life Sciences, Pocheon-si, South Korea). Cells were washed with PBS and plated in Dulbecco's modified Eagle's medium-high glucose (DMEM-HG, Welgene) containing 10% FBS, 100 μg/mL streptomycin, and 100 U/mL penicillin. After 48 h, the medium was replaced to remove non-adherent cells. CAFs were then expanded for three weeks [38].

## Flow cytometry

CAFs were stained with anti-fibroblast activation protein (FAP) antibody (#427819, mouse monoclonal $IgG_1$, Novus Biologicals, Littleton, CO, USA) followed by fluorescein isothiocyanate (FITC)-conjugated anti-mouse $IgG_1$ antibody. Jurkat and THP-1 cells were incubated with FITC-conjugated anti-human CCR1 (5F10B29, mouse monoclonal), allophycocyanin (APC)-conjugated anti-human CCR2 (K036C2, mouse monoclonal), and APC-conjugated anti-human CCR3 (5E8, mouse monoclonal) antibodies (all from BioLegend, San Diego, CA, USA) for 30 min on ice. Isotype antibody-stained cells were used as controls. Cells were centrifuged at $400 \times g$ for 5 min at room temperature and fixed with 1% paraformaldehyde in flow cytometry buffer (0.5% FBS in PBS). Cells were detected using a NovoCyte flow cytometer (ACEA Biosciences, San Diego, CA, USA) and analyzed with NovoExpress software (ACEA Biosciences).

## Transwell migration assay

Transwell migration assay was performed using Transwell®-24 well permeable support plates with an 8.0-μm pore size polycarbonate membrane (Corning, Corning, NY, USA). CAFs ($10^5$ cells/well) were cultured in the lower chamber for 24 h. For the migration assay, lower cells were treated with 10 μg/mL CCL7-blocking antibody (R&D Systems, Inc., Minneapolis, MN, USA) and/or 100 μg/mL exosomes from FBS, HT-29 cells, or SW480 cells. THP-1 cells ($2 \times 10^5$ in each well) were seeded in the upper chambers in 100 μL serum-free medium. The chamber was incubated at 37˚C for 5 h. RPMI1640 supplemented with 10% FBS was added to the lower chamber as a positive control for the migration of THP-1 cells, and serum free-RPMI-1640 was used as a negative control. Migrated THP-1 cells in lower chambers were counted after trypan blue staining using a hemocytometer.

## Small RNA library construction and sequencing analysis

For miRNA sequencing, total RNA was isolated using an extraction kit (Qiagen, Germany) and sent to Macrogen (Seoul, South Korea) for small RNA library construction and sequencing. The RNA isolated from each sample was used to construct sequencing libraries with the SMARTer® smRNA-Seq Kit from Illumina, following the manufacturer's protocol. The libraries were pooled in equimolar amounts and sequenced on an Illumina HiSeq 2500 (Illumina, USA) instrument to generate 101 base reads. Image decomposition and quality value calculations were performed using the modules of the Illumina pipeline. Raw data (the reads for each miRNA) were normalized by relative log expression normalization using DESeq2. miRNA target genes were predicted using TargetScan (http://www.targetscan.org/vert_72/), miRDB (http://mirdb.org/), and microRNA.org (http://www.microrna.org/) databases.

## Quantitative reverse transcription (qRT)-polymerase chain reaction (PCR) of exosomal miRNAs

qRT-PCR was performed on the miRNA from exosomes to validate the miRNA data. To harvest cell exosomes for qRT-PCR, $4 \times 10^6$ HT-29 and SW480 cells were cultured in RPMI 1640 with 10% exosome-free FBS for 24 h in a humid atmosphere with 5% $CO_2$ at 37°C. Each cell supernatant was centrifuged for 5 min at 1,300 rpm and filtered using a 0.22-μm syringe filter. The miRNA of exosomes was extracted using the Exo2D™-EV isolation kit for RNA analysis (EXO-SOMEplus, Suwon-si, South Korea). Samples were centrifuged at $3000 \times g$ for 15 min to remove the cells and debris, and the supernatant was transferred to a new tube. Exo2D™ was incubated at 37°C for 15 min and inverted. The samples were shaken every 5 min to keep the samples optically opaque during the process. Next, 10 mL of sample per 2 mL of Exo2D™ reagent B was added and mixed by inverting. The mixtures were centrifuged at $3,000 \times g$ for 30 min at 4°C. The aqueous phase of Exo2D™ grabs exosomes was precipitated. Exosomes were dissolved in a small volume of the aqueous phase. The phase appeared as a white pellet, and the remainder was eliminated. The homogenized samples were resuspended in 100 μL of PBS. miRNAs are not naturally polyadenylated. With the MystiCq microRNA cDNA synthesis mix (Merck, Darmstadt, Germany), miRNAs were polyadenylated through a poly(A) polymerase reaction and subsequently added to convert the poly(A) tailed microRNAs into cDNA using an oligo-dT adapter primer. The adapter primer incorporates a unique sequence at its 5' end, which allows for the amplification of cDNAs in real-time RT-qPCR reactions. MystiCq universal PCR primers, miRNA primers has-miR-1246 (**AAUGGAUUUUUGGAGCAGG**), has-miR-367-3P (**AAUUGCACU UUAGCAAUGGUGA**), hsa-let-7D-5P (**AGAGGUAGUAGGUUGCAUAGUU**), and SNORD48 (human positive control primer, AGUGAUGAUGACCCCAGGUAACUCUGAGUGUGUCGCUGAUG CCAUCACCGCAGCGCUCU−GACC) were purchased from Merck. Real-time PCR was conducted to quantify the expression of specific genes using a KAPA SYBR® FAST qPCR kit (KAPA Biosystems Inc., Woburn, MA, USA) with an ABI PRISM 7000 sequence detection system (Applied Biosystems, Foster City, CA, USA). The expression of miR-6126, miR-367-3p, let-7d-5p, and miR-1246 was determined relative to that of SNORD 48, and levels were calculated using the $2^{-\Delta\Delta CT}$ method.

## Statistical analysis

Statistical analysis was performed using GraphPad Prism version 6.04 (GraphPad Software Inc., San Diego, CA, USA). All data are presented as mean ± SEM. Statistical significance was determined by one-way analysis of variance (ANOVA) in conjunction with Dunnett's *post hoc* test as applied to the cell migration assay. All analyses were performed using Prism 8 (GraphPad Software, Inc., La Jolla, CA, USA). A *P*-value of $<0.05$ was considered statistically significant.

## Results

### Exosome isolation from HT29 and SW480 cell lines

Two CRC cell lines, HT-29 and SW480, were cultured in exosome-free FBS-containing media for 24 or 48 h (Fig 1A). Exosomes were isolated and purified from culture supernatants. The cup-shaped structures and sizes were identified by electron microscopy (Fig 1B). The presence of CD9 and CD63 was confirmed by immunoblotting (Fig 1C). The particle size distribution and concentration of isolated exosomes were analyzed by Nanosight particle tracking analysis (Fig 1D).

### CAF isolation

To isolate CAFs from CRC, surgically removed tissue from the tumor was minced and enzyme-digested. CAFs were typically spindle-shaped (Fig 2A), and the expression of FAP was observed in this batch of cells (Fig 2B).

### Migration of THP-1 cells was affected by CCL7 from HT29 and SW480 exosome-treated CAFs

Cell migration assays were performed with CAFs incubated with exosomes from HT29 and SW480 for 5 h to evaluate the effect of CRC cell-derived exosomes on the chemotactic effect in the tumor microenvironment. First, the expression levels of CCR1, CCR2, and CCR3 in Jurkat and THP-1 cells were analyzed via flow cytometry. It was found that there was no expression of CCR1, CCR2, and CCR3 in Jurkat cells, while CCR2 was highly expressed in THP-1 cells (Fig 3A). A Transwell assay was performed with CAFs only, CAFs and CCL7-blocking antibodies, CAFs and HT29 exosomes, and CAFs and HT480 exosomes. It was found that THP-1 cells migrated toward CAFs and decreased by CCL7-blocking antibody treatment. It was found that migration of THP-1 cells was decreased in HT29 and SW480 exosome-treated CAFs rather than CAFs-only conditions (Fig 3B).

### Sequencing analysis of miRNAs in exosomes

Exosomes were isolated, and miRNA sequencing was performed to compare the miRNAs in exosomes from those in HT-29 and SW480 cells. The miRNA reads of samples were lower 48

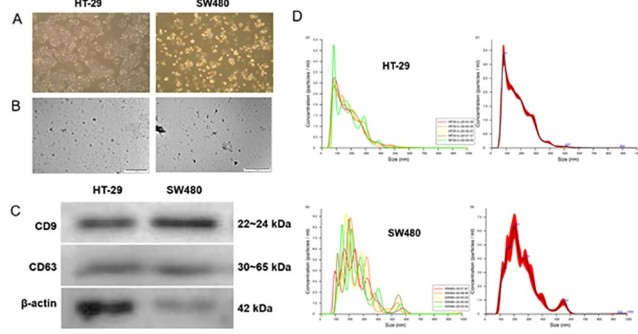

**Fig 1. Exosomes isolated from HT-29 and SW480 cell lines.** (A) Cell morphology of HT-29 and SW480 cells in RPMI-1640 with exosome-free 10% FBS (original magnification ×100). (B) Exosomes secreted from HT-29 and SW480 cells were detected by electron microscopy (scale bar, 1 um). (C) Immunoblotting assay of CD9 (22~24 kDa) and CD63 (30~65 kDa) in exosomes from HT-29 and SW480 cells. Beta-actin (42 kDa) blot was used as the loading control. (D) Exosomes from HT-29 and SW480 cells were detected by Nanosight particle tracking analysis. The left column represents batch-to-batch variation, and the right column shows the overall size of the exosomes isolated from HT-29 and SW480, respectively.

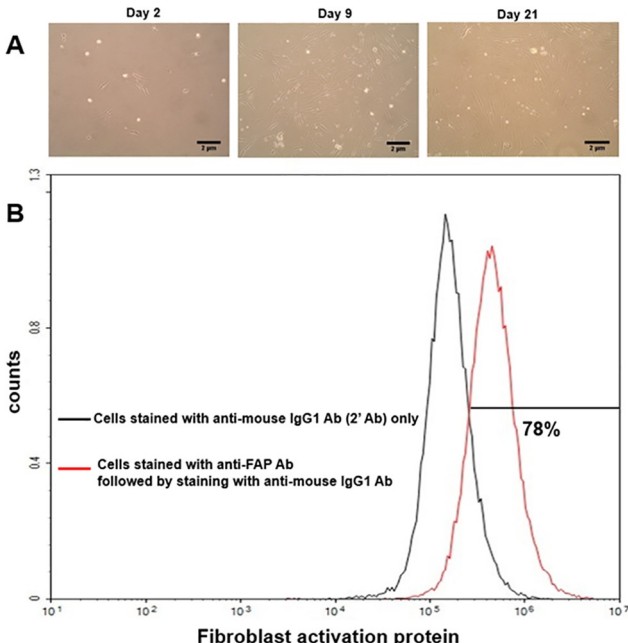

**Fig 2. CAF isolation from a patient with colorectal cancer.** (A) Isolated CAFs are spindle-shaped. Adherent cells can be observed after day 2 (original magnifications are x100 and 2μm scale bars were inserted). (B) The expression level of fibroblast activation protein was analyzed with flow cytometry.

h after medium replacement than those 24 h after medium replacement. The 14 miRNAs that were most highly expressed were selected and are listed in Table 2. Thereafter, miRNA target genes were predicted using sequence-based database tools, such as TargetScan. Possible target genes of the 14 miRNAs from Table 2 were selected. It was observed that CCL7-targeting let-7d-5p was detected in exosomes from HT-29 and SW480 cells.

## qRT-PCR for miR-6126, miR-1246, miR-367-3p, and let-7d-5p in exosomes

qRT-PCR was performed for miR-6126, miR-1246, miR-367-3p, and let-7d-5p to confirm the miRNA read count in the sequencing analysis. It was found that miR-6126, miR-1246, miR-367-3p, and let-7d-5p were expressed along with SNORD 48 (control) (Fig 4).

## Discussion

The tumor microenvironment plays a significant role in tumor occurrence and development, and cell-to-cell communication is an essential mechanism in the tumor microenvironment, which maintains tissue homeostasis and normal cellular activities [39, 40]. The mechanisms of cell-to-cell communication are direct cell-to-cell contact mediated by integral membrane proteins, indirect contact through the extracellular matrix, and distant communication by circulating miRNA via exosomes or extracellular miRNA in the extracellular microenvironment [41]. Recently, the role of tumor cell-derived exosomes in cell-to-cell communication in the tumor microenvironment has drawn attention, and their pathogenesis has been gradually revealed.

Exosomes are 50–120 nm cup-shaped vesicles that were discovered in the early 1980s by Pan and Johnstone [42, 43]. The biogenesis of exosomes initiates in the endosomal system and several steps are involved [8]. After manufacturing endosomes through inward budding of

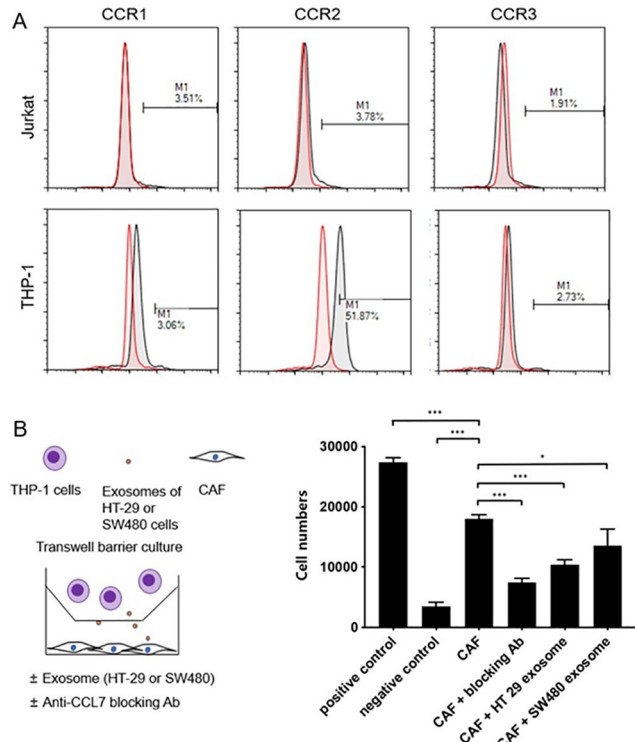

**Fig 3. Transwell migration assay.** (A) The expression levels of CCR1, CCR2, and CCR3 in Jurkat and THP-1 cells were analyzed via flow cytometry. (B) Schematic presentation of the cell migration assay toward CAFs. CAFs were placed in the lower chamber with exosomes from FBS, HT-29 cells, and SW480 cells, and THP-1 cells were placed in the upper chamber. Serum-free media were used as negative control, and 10% FBS-containing media were used as positive control. After 5 h, THP-1 cells in the lower chamber were collected and counted. Data are expressed as the mean +/- SEM and were analyzed by one-way ANOVA ($^*P < 0.05$, $^{**}P < 0.01$, and $^{***}P < 0.001$).

**Table 2. List of miRNAs highly expressed in exosomes from HT-29 and SW480 cells.**

| miRBase_Link | HT-29_24hr | HT-29_48hr | SW480_24hr | SW480_48hr |
|---|---|---|---|---|
| hsa-let-7a-5p | 1667 | 224 | 3110 | 1207 |
| hsa-let-7b-5p | 3231 | 435 | 5937 | 1944 |
| hsa-let-7d-5p | 122 | 27 | 170 | 58 |
| hsa-let-7e-5p | 339 | 63 | 762 | 208 |
| hsa-let-7f-5p | 717 | 69 | 1348 | 358 |
| hsa-let-7g-5p | 115 | 44 | 211 | 88 |
| hsa-miR-1246 | 1561 | 392 | 4337 | 2026 |
| hsa-miR-1290 | 1022 | 302 | 2522 | 3031 |
| hsa-miR-185-5p | 72 | 5 | 166 | 24 |
| hsa-miR-191-5p | 110 | 34 | 230 | 80 |
| hsa-miR-23a-3p | 105 | 80 | 198 | 174 |
| hsa-miR-423-5p | 300 | 148 | 556 | 258 |
| hsa-miR-6126 | 77 | 5 | 211 | 42 |
| hsa-miR-92a-3p | 119 | 27 | 169 | 110 |

Numbers represent the normalized detection reads of the miRNA sequence.

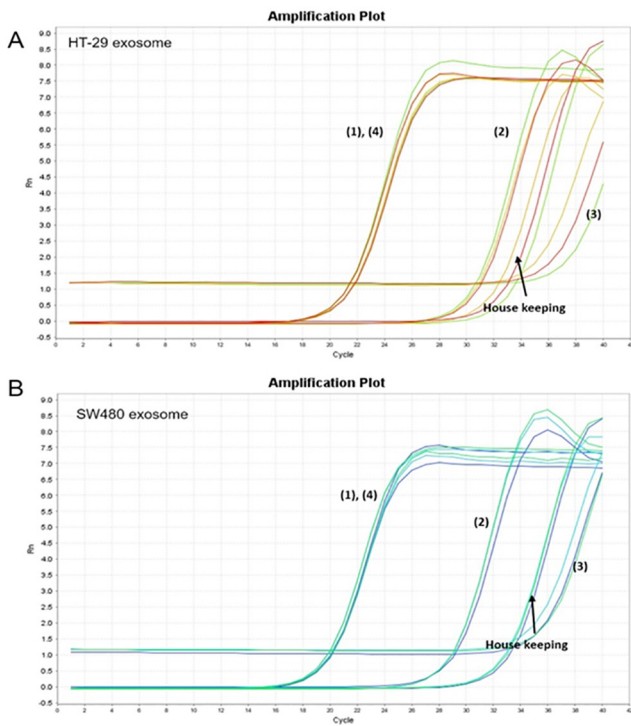

**Fig 4. qRT-PCR for miRNA validation in exosomes.** (A) HT-29 cell- and (B) SW480 cell-derived exosome analysis. (1) miR-6126, (2) miR-367-3p, (3) let-7d-5p, (4) miR-1246, and housekeeping RNA (SNORD 48).

clathrin-coated regions of the cell membrane, the membrane of endosomes bulges inward. Numerous small vesicles containing cytoplasmic contents appear in endosomes, which are multivesicular bodies [44]. These multivesicular bodies secrete their intraluminal vesicles by fusing with the plasma membrane into extracellular spaces, and secreted vesicles into extracellular spaces are called exosomes [45–47]. During this process, exosomes receive the components of originating cells and can carry various types of molecules, such as miRNA, mRNA, DNA, long noncoding RNA, proteins, and lipids, based on their sources [10, 11]. The exosomes from various organisms have been found to include 194 lipids, 4,563 proteins, 1,639 mRNAs, and 764 miRNAs, based on the database ExoCarta (http://www.exocarta.org/) [48]. Exosomes are composed of a common combination of protein and lipid components irrespective of their origin [49]. The membrane of exosomes is composed of high levels of cholesterol, ceramide, sphingomyelin, and glycerophospholipids, with long and saturated fatty-acyl chains [50]. Furthermore, exosomal proteins include tetrapenins (CD9, CD63, CD81, and CD82), heat shock proteins (HSP60, HSP70, and HSP90), cytoskeletal proteins (myosin, β-actin, and tubulin), multivesticular body-related proteins (Alix and Tsg101), integrins, fusion proteins (annexins, Rab GTPases, flotillins), and glycolytic proteins (enolase 1 and glyceraldehyde 3-phosphate dehydrogenase) [51, 52]. In this study, exosomes were isolated from cultured colon cancer cell lines HT-29 and SW480. These were confirmed as exosomes based on their surface markers CD9 and CD63 and size measurement in the Transmission Electron Microscopy (TEM) and Nanosight particle tracking analysis (Fig 1). Exosomes were isolated by classical ultracentrifugation and precipitation. Higher exosome yield was reported in the precipitation than in the ultracentrifugation, and the yield of exosome isolation was superior in the precipitation [53].

The development and progression of cancer depend not only on its own characteristics but also the surrounding tumor microenvironment, which is composed of various types of cells including tumor cells and stromal cells, such as fibroblasts, adipocytes, endothelial cells, immune cells, and mesenchymal stem cells [54]. Exosomal miRNAs play a significant role in the tumor microenvironment as an intercellular signaling molecule that mediates cancer progression and suppression [55]. Moreover, they influence stromal components and the immune system [56]. In this study, we focused on CAFs as a target of tumor-derived exosomal miRNAs among the constituents of the tumor microenvironment. CAFs are vital components of the tumor microenvironment that interact with cancer cells and play a significant role in mediating their formation and activation [6, 57–59]. CAFs are morphologically spindle-shaped and characterized by expression of α-smooth muscle actin, FAP-α, fibroblast specific protein-1, platelet-derived growth factor receptor-α, and -β [38]. Here, CAFs were prepared by enzyme digestion followed by the elimination of floating cells after 48 h, while maintaining adherent cells expressing FAP-α (Fig 2A). The huddle of CAF isolation from CRC, however, contamination is suspected at the source or during cell preparation. These cell lines should be handled alone, preferably in quarantine.

CAFs secrete growth factors and cytokines such as CXCL12 (stromal cell-derived factor-1), CCL7 (monocyte-chemotactic protein 3), transforming growth factor-beta, fibroblast growth factors, hepatocyte growth factor, periostin, and tenascin C, which stimulate cancer cells to enhance survival, proliferation, stemness, metastasis, and resistance to therapy [60–67]. Among the cytokines secreted by CAFs, CCL7 is known to be tumorigenic, which may promote tumor growth, invasion, and metastasis. CCL7 is associated with recruiting inflammatory cells of monocytes, macrophages, and myeloid-derived suppressor cells, which promote the development of type 2 macrophages that inhibit antitumor immune response, thereby allowing tumor progression [68]. Furthermore, the recruitment of inflammatory cells results in increased vascular permeability that favors cancer metastasis [69–71]. According to previous studies, higher CCL7 expression has been identified in metastatic renal cell carcinoma than in primary renal cell carcinoma and is upregulated in lung adenomas, which show marker accumulation of immune cells [71, 72]. In CRC, overexpression of CCL7 was also associated with cancer proliferation, invasion, and migration *in vitro* and *in vivo* [73, 74]. Here, exosomes from the CRC cell lines HT-29 and SW480 affected CCL7 secretion from CAFs and interfered with the migration of CCR2+ monocytic THP-1 cells *in vitro* (Fig 3B). It can be speculated that the secretion of immune cell attractants by CAFs was suppressed by exosomes derived from cancer cells. In our subsequent experiments, miRNAs of exosomes from CRC cell lines, HT-29 and SW480, were isolated and compared. The miRNA sequencing and miRNA validation results did not show significant differences between these two cells, and both cell lines contained let-7d in secreted exosomes. According to previous reports, CCL7 is known to be downregulated by miRNA let-7d, which is one of the miRNAs expressed by exosomal miRNA from the human colon cancer cell line sequencing in our experiment (Table 2 and Fig 4). As a tumor suppressor, let-7d specifically binds to the 3'-UTR of CCL7 mRNA and modulates its expression in a negative feedback manner, which is frequently downregulated in many human malignancies, such as lung cancer, breast cancer, and hepatocellular carcinoma [75–78].

In conclusion, exosomes from the CRC cell lines HT-29 and SW480 affected CCL7 secretion from CAFs, possibly via the miRNA let-7d, and interfered with the migration of CCR2 + monocytic THP-1 cells *in vitro*. The present study suggests a role of exosomal miRNAs in the tumorigenesis of colon cancer by affecting the tumor microenvironment. The overall effect of exosomes on CAFs in RNA sequencing requires further comparison. Further validation of the immune cell migration and tumorigenesis effect of exosomes *in vivo* is also needed.

## Supporting information

**S1 Raw images.**
(TIF)

## Author Contributions

**Conceptualization:** Gyoung Tae Noh, So-Youn Woo, Bo-Young Oh, Kang Young Lee, Ryung-Ah Lee.

**Formal analysis:** Gyoung Tae Noh.

**Investigation:** Gyoung Tae Noh, Kyung-Ah Cho, So-Youn Woo, Kang Young Lee.

**Methodology:** Gyoung Tae Noh, Jiyun Kwon, Jungwoo Kim, Minhwa Park, Da-Won Choi, Kyung-Ah Cho, So-Youn Woo, Ryung-Ah Lee.

**Resources:** Jiyun Kwon, Jungwoo Kim, Minhwa Park, Da-Won Choi, Kyung-Ah Cho, So-Youn Woo.

**Supervision:** Bo-Young Oh, Ryung-Ah Lee.

**Validation:** Gyoung Tae Noh, Bo-Young Oh, Kang Young Lee.

**Writing – original draft:** Gyoung Tae Noh.

**Writing – review & editing:** Ryung-Ah Lee.

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
