## [Decision Letter · Decision Letter 0]

6 Oct 2020

PONE-D-20-27371

Verification of the role of exosomal microRNA in tumorigenesis for colorectal cancer using human colorectal cancer cell lines

PLOS ONE

Dear Dr. Lee,

Thank you for submitting your manuscript to PLOS ONE. After careful consideration, we feel that it has merit but does not fully meet PLOS ONE’s publication criteria as it currently stands. Therefore, we invite you to submit a revised version of the manuscript that addresses the points raised during the review process.

Please respond to all critique, point-by-point. In particular:

- The manuscript would greatly profit from a language checkup

- page 14 and table 1: provide information on the sample size

- The literature is in part outdated. One may consider some of the newer literature suggested by referee 1.

We look forward to receiving your revised manuscript.

Kind regards,

Klaus Roemer

Academic Editor

PLOS ONE

Journal Requirements:

3. Thank you for submitting the above manuscript to PLOS ONE. During our internal evaluation of the manuscript, we found significant text overlap between your submission and the following previously published works, some of which you are an author.

https://onlinelibrary.wiley.com/doi/abs/10.1002/jcp.26481

https://dmm.biologists.org/content/dmm/11/4/dmm029447.full.pdf?rss=1

https://molecular-cancer.biomedcentral.com/articles/10.1186/s12943-018-0897-7

https://peerj.com/articles/4928.pdf

Please revise the manuscript to rephrase the duplicated text, cite your sources, and provide details as to how the current manuscript advances on previous work. Please note that further consideration is dependent on the submission of a manuscript that addresses these concerns about the overlap in text with published work.

4. At this time, we ask that you please provide scale bars on the microscopy images presented in Figure 2 and refer to the scale bar in the corresponding Figure legend.

5. Please provide additional information about each of the cell lines used in this work, including any quality control testing procedures (authentication, characterisation, and mycoplasma testing).

For more information, please see http://journals.plos.org/plosone/s/submission-guidelines#loc-cell-lines

6. In your Methods section, please provide additional information about the tissue specimens used in this study, the method used to collect them, and the demographic details of the patients from which they were collected. Please ensure you have provided sufficient details to replicate the analyses such as:

a) the date range (month and year) during which you collected specimens, 

b) a description of how participants were recruited to provide samples, and

c) eligibility criteria for being included in this part of the study.

7. Thank you for including your ethics statement in the Manuscript methods:

'Tumor samples from patients undergoing surgery were obtained at Ewha University Medical Center (Seoul, Korea) in accordance with the ethical guidelines of the institutional review board (IRB No. SEUMC 2019-12-028). All patients provided their formal, informed, and written consent, agreeing to supply a biopsy for this study.

CAFs were isolated from the tumor samples of patients.'

a. Please amend your current ethics statement to include the full name of the ethics committee/institutional review board(s) that approved your specific study and confirm that your named institutional review board or ethics committee specifically approved this study.

8. PLOS ONE now requires that authors provide the original uncropped and unadjusted images underlying all blot or gel results reported in a submission’s figures or Supporting Information files. This policy and the journal’s other requirements for blot/gel reporting and figure preparation are described in detail at https://journals.plos.org/plosone/s/figures#loc-blot-and-gel-reporting-requirements and https://journals.plos.org/plosone/s/figures#loc-preparing-figures-from-image-files. When you submit your revised manuscript, please ensure that your figures adhere fully to these guidelines and provide the original underlying images for all blot or gel data reported in your submission. See the following link for instructions on providing the original image data: https://journals.plos.org/plosone/s/figures#loc-original-images-for-blots-and-gels.

9. Your ethics statement should only appear in the Methods section of your manuscript. If your ethics statement is written in any section besides the Methods, please delete it from any other section.

Reviewers' comments:

Reviewer's Responses to Questions

**Comments to the Author**

1. Is the manuscript technically sound, and do the data support the conclusions?

Reviewer #1: Yes

Reviewer #2: Yes

2. Has the statistical analysis been performed appropriately and rigorously? 

Reviewer #1: Yes

Reviewer #2: Yes

3. Have the authors made all data underlying the findings in their manuscript fully available?

Reviewer #1: Yes

Reviewer #2: Yes

4. Is the manuscript presented in an intelligible fashion and written in standard English?

Reviewer #1: No

Reviewer #2: Yes

5. Review Comments to the Author

Reviewer #1: This is very attractive work.

Some minor comments:

1- Needs some language corrections

2- Add size marker to WB gels

3-Add the below references:

For CRC:

a-Epigenomics. 2019 Nov;11(14):1627-1645.

b-Lancet Gastroenterol Hepatol. 2019 Dec;4(12):913-933.

c-Pharmacol Res. 2020 Aug 18;161:105133.

d-Crit Rev Oncol Hematol. 2020 Jan;145:102854

For Exosome and Exosomal microRNA

a-Cell Commun Signal. 2020 Sep 11;18(1):149.

b-Cell Commun Signal. 2020 Aug 3;18(1):120.

c-Mol Ther Nucleic Acids. 2020 Sep 4;21:51-74.

d-Epigenomics. 2020 Feb;12(4):353-370.

4- Check primers sequences

Reviewer #2: This manuscript mainly about analyze the effect of exosomal miRNA on the tumor microenvironment, monocytic cell line THP-1 migration was evaluated by Transwell migration assay with CAFs isolated from colon cancer patients.,which contribute to providing new insights colorectal cancer. This article has obvious clinical value and scientific significance.

This paper is written smoothly with clear thinking and detailed

experimental methods, but there are some minor problems:

1. Page12，the line10 and 12 of the “Introduction”, Tumor-derived and cancer-derived should be unified.

2. Page13, the last setence of the 2nd paragraph, “When cancer arises in the adult organ, the dominant niche likely includes the expansion of quiescent fibroblasts residing in the host tissue in response to the injury caused by the developing neoplasm.” The difference between cancer and neoplasm? It may better to unify the same noun in the whole article.

3. Page13, “Cancer-associated fibroblasts (CAFs) are vital constituents of the tumor microenvironment”(the line5) and “CAFs which are vital constituents of the tumor microenvironment.”(the line16) may repeat?

4. Page14, What is the sample size? It should be specified, and other relevant information of the sample should also be specified in the Table1.

5. The Figure can be clearer? Especially Figure 1C and Figure 2B.

6. Some of references are old, it is better to learn from the latest views.

Overall, I think this article has certain innovation, but there are also some problems. I think this article needs to be reviewed again after revision.

6. PLOS authors have the option to publish the peer review history of their article (what does this mean?). If published, this will include your full peer review and any attached files.

Reviewer #1: No

Reviewer #2: No

---

## [Author Response · Author response to Decision Letter 0]

15 Oct 2020

Response to reviewers

General statement

1. The manuscript would greatly profit from a language checkup

This manuscript underwent language editing by a professional scientific editing service (Editage) according to your recommendation.

2. page 14 and table 1: provide information on the sample size

The colorectal cancer cells used in the experiment were not collected from patients, but cell lines that had been used in various existing studies manufactured by accredited institutions were purchased and cultured. Therefore, we think that it is not important to specify the sample size.

3. The literature is in part outdated. One may consider some of the newer literature suggested by referee 1.

In accordance with the recommendations, the introduction and discussion were revised by citing the latest references, including those suggested by the reviewer 1.

Journal requirement

The article was revised according to the proposed format.

This manuscript underwent language editing by a professional scientific editing service (Editage) according to your recommendation.

3. During our internal evaluation of the manuscript, we found significant text overlap between your submission and the following previously published works, some of which you are an author. Please revise the manuscript to rephrase the duplicated text, cite your sources, and provide details as to how the current manuscript advances on previous work.

To avoid duplication, the text and references of the manuscript were extensively modified as recommended.

4. At this time, we ask that you please provide scale bars on the microscopy images presented in Figure 2 and refer to the scale bar in the corresponding Figure legend.

As pointed out, scale bars were inserted in the figure and added to the figure legend.

5. Please provide additional information about each of the cell lines used in this work, including any quality control testing procedures (authentication, characterisation, and mycoplasma testing).

The points pointed out are added to the text as follows.

“All the cell lines were regularly tested using quantitative PCR, MycoTOOL PCR Mycoplasma Detection Kit (Roche Custom Biotech, Mannheim, Germany). The passage number of HT-29 cells and SW480 cells at purchase were P130 (#lot 700190500 and P99 (#lot 70013709), respectively.”

6. In your Methods section, please provide additional information about the tissue specimens used in this study, the method used to collect them, and the demographic details of the patients from which they were collected. Please ensure you have provided sufficient details to replicate the analyses such as:

a) the date range (month and year) during which you collected specimens, 

b) a description of how participants were recruited to provide samples, and

c) eligibility criteria for being included in this part of the study.

As recommended, information on tissue specimen sampling was added as follows.

“Between June 2020 and August 2020, two patients underwent surgical resection for colon cancer were enrolled in this study. The eligibility criteria were a histologically confirmed colonic or rectal adenocarcinoma and major resection of primary lesion. Patients who underwent preoperative treatment such as chemotherapy, radiotherapy, and endoscopic resection were excluded. Before surgery, patients were given an explanation of the purpose and risk of this study and decided to consent to tissue collection. All patients provided their formal, informed, and written consent, agreeing to supply a biopsy for this study.” 

7. Thank you for including your ethics statement in the Manuscript methods:

a. Please amend your current ethics statement to include the full name of the ethics committee/institutional review board(s) that approved your specific study and confirm that your named institutional review board or ethics committee specifically approved this study.

The IRB information was added as follows.

“The study and informed consent were reviewed and approved by the institutional review board of Ewha Medical Center Seoul hospital. (IRB No. SEUMC 2019-12-028).”

The above contents were added to the submission form.

8. PLOS ONE now requires that authors provide the original uncropped and unadjusted images underlying all blot or gel results reported in a submission’s figures or Supporting Information files.

In your cover letter, please note whether your blot/gel image data are in Supporting Information or posted at a public data repository, provide the repository URL if relevant, and provide specific details as to which raw blot/gel images, if any, are not available.

We uploaded the original blot image data in Supporting Information according to the journal’s guideline and noted it in the cover letter.

9. Your ethics statement should only appear in the Methods section of your manuscript. If your ethics statement is written in any section besides the Methods, please delete it from any other section.

As pointed out, we deleted the ethics statement outside of the methods section.

Reviewer comments to the author

Reviewer #1

1. Needs some language corrections

As pointed out, this manuscript underwent language editing by a professional scientific editing service (Editage) according to your recommendation.

2. Add size marker to WB gels

The image in Figure 1c was obtained by capturing the original image and cannot display a size marker, but a description for this was added. The original blot image with size markers is attached.

3. Add the below references

The introduction and discussion were revised by adding the suggested references.

4. Check primers sequences

As pointed out, We added the sequence of miRNA PCR primers for qRT-PCR of exosomal miRNAs as follows.

“MystiCq universal PCR primers, miRNA primers gas-MiR-1246 (AAUGGAUUUUUGGAGCAGG), gas-MiR-367-3P (AAUUGCACUUUAGCAAUGGUGA), has-let-7D-5P (AGAGGUAGUAGGUUGCAUAGUU), and SNORD48 (human positive control primer, AGUGAUGAUGACCCCAGGUAACUCUGAGUGUGUCGCUGAUGCCAUCACCGCAGCGCUCUGACC) were purchased from Merck.”

Reviewer #2

1. Page12，the line10 and 12 of the “Introduction”, Tumor-derived and cancer-derived should be unified.

As pointed out, the terms were unified and reflected in the text.

2. Page13, the last sentence of the 2nd paragraph, “When cancer arises in the adult organ, the dominant niche likely includes the expansion of quiescent fibroblasts residing in the host tissue in response to the injury caused by the developing neoplasm.” The difference between cancer and neoplasm? It may better to unify the same noun in the whole article.

As pointed out, the terms were unified and reflected in the text.

3. Page13, “Cancer-associated fibroblasts (CAFs) are vital constituents of the tumor microenvironment” (the line5) and “CAFs which are vital constituents of the tumor microenvironment.” (the line16) may repeat?

The revision of the item was reflected in the text.

4. Page14, What is the sample size? It should be specified, and other relevant information of the sample should also be specified in the Table1.

Colorectal cancer cells used in the experiment were not collected from patients, but cell lines that had been used in various existing studies manufactured by accredited institutions were purchased and cultured. Therefore, we think that it is not considered important to specify the sample size.

5. The Figure can be clearer? Especially Figure 1C and Figure 2B.

We replaced the Figure 1C (and provide raw data as well) and Figure 2B.

6. Some of references are old, it is better to learn from the latest views.

As pointed out, the bibliography of the overall article was revised to the latest reference and reflected in the text.

---

## [Editor Report · Decision Letter 1]

27 Oct 2020

Verification of the role of exosomal microRNA in colorectal tumorigenesis using human colorectal cancer cell lines

PONE-D-20-27371R1

Dear Dr. Lee,

We’re pleased to inform you that your manuscript has been judged scientifically suitable for publication and will be formally accepted for publication once it meets all outstanding technical requirements.

Kind regards,

Klaus Roemer

Academic Editor

PLOS ONE
---

## [Editor Report · Acceptance letter]

29 Oct 2020

PONE-D-20-27371R1 

Verification of the role of exosomal microRNA in colorectal tumorigenesis using human colorectal cancer cell lines 

Dear Dr. Lee:

I'm pleased to inform you that your manuscript has been deemed suitable for publication in PLOS ONE. Congratulations! Your manuscript is now with our production department. 

Kind regards, 

on behalf of

Dr. Klaus Roemer 

Academic Editor

PLOS ONE